# Community health worker–facilitated telehealth for moderate–severe hypertension care in Kenya and Uganda: A randomized controlled trial

Matthew D. Hickey[1]*, Asiphas Owaraganise[2], Sabina Ogachi[3], Norton Sang[3], Erick M. Wafula[3], Jane Kabami[2], Nicole Sutter[1], Jennifer Temple[1], Anthony Muiru[4], Gabriel Chamie[1], Elijah Kakande[2], Maya L. Petersen[5], Laura B. Balzer[5], Diane V. Havlir[1], Moses R. Kamya[2,6], James Ayieko[3]

1 Division of HIV, Infectious Disease, & Global Medicine, University of California San Francisco, San Francisco, California, United States of America, 2 Infectious Diseases Research Collaboration, Kampala, Uganda, 3 Kenya Medical Research Institute, Nairobi, Kenya, 4 Division of Nephrology, University of California San Francisco, San Francisco, California, United States of America, 5 School of Public Health, University of California Berkeley, Berkeley, California, United States of America, 6 School of Medicine, Makerere University, Kampala, Uganda

* matt.hickey@ucsf.edu

## Abstract

### Background

Hypertension is underdiagnosed and undertreated in sub-Saharan Africa. Improving hypertension treatment within primary health centers can improve cardiovascular disease outcomes; however, individuals with moderate–severe hypertension face additional barriers to care, including the need for frequent clinic visits to titrate medications. We conducted a pilot study to test whether a clinician-driven, community health worker (CHW)–facilitated telehealth intervention would improve hypertension control among adults with severe hypertension in rural Uganda and Kenya.

### Methods and findings

We conducted a pilot randomized controlled trial (RCT) of hypertension treatment delivered via telehealth by a clinician (adherence assessment, counseling, decision-making) and facilitated by a CHW in the participant's home, compared to clinic-based hypertension care (NCT04810650). We recruited adults ≥40 years with BP ≥ 160/100 mmHg at household screening by CHWs, with no restrictions by HIV status. After initial evaluation at the clinic, participants were randomized to telehealth or clinic-based hypertension follow-up. Randomization assignment was not blinded, except for the study statistician. All participants were treated using standard country guideline-based antihypertensive drugs. The primary outcome was hypertension control at 24 weeks (BP < 140/90 mmHg). We also assessed hypertension control at 48 weeks. In intention-to-treat analyses, we compared outcomes between randomized

**Data availability statement:** All relevant data are within the manuscript and its Supporting Information files. The code used in the analysis is available from Github https://github.com/LauraBalzer/search_severe_htn.

**Funding:** This work was supported by the National Institute of Allergy and Infectious Diseases, the National Heart, Lung, and Blood Institute, the National Institute of Mental Health, and the National Institute on Alcohol Abuse and Alcoholism (U01-AI150510 to DVH, MLP, and MRK) and the National Heart, Lung, and Blood Institute (K23HL162578 to MDH). The funders had no role in study design, data collection and analysis, decision to publish, or preparation of the manuscript.

**Competing interests:** The authors have declared that no competing interests exist.

arms with targeted minimum loss-based estimation using sample-splitting to select optimal adjustment covariates (candidates: age, sex, baseline hypertension severity, and country). We screened 2,965 adults ≥40 years, identifying 266 (9%) with severe hypertension and enrolling 200 (98 telehealth arms, 102 clinic arms). Participants were 67% women, median age of 62 years (Q1–Q3 51–72); 14% with HIV. Week 24 blood pressure was measured in 96/99 intervention and 99/102 control participants; week 24 hypertension control was 77% in telehealth and 51% in clinic arms (risk difference (RD) 26%, 95% confidence interval (CI) [14%, 38%], p < 0.001). Week 48 hypertension control was 86% in telehealth and 44% in clinic arms (RD 42%, 95% CI [30%, 53%], p < 0.001). Three participants died (telehealth: 2, clinic: 1); all deaths were unrelated to the study interventions. Our study was limited by its small sample size, although findings are strengthened by being conducted in three primary health centers across two countries.

## Conclusion

In this pilot, RCT, clinician-driven, CHW-facilitated telehealth for hypertension management improved hypertension control and reduced severe hypertension compared to clinic-based care. Telehealth focused on individuals with moderate–severe hypertension is a promising approach to improve outcomes among those with the highest risk for CVD.

---

Author summary

### Why was this study done?

- Fewer than 10% of people with hypertension in Africa have controlled blood pressure, contributing to an increasing burden of cardiovascular disease, such as heart attacks and strokes.

- Previous studies have shown that hypertension can successfully be treated in primary care clinics; however, retention in care and hypertension control remain sub-optimal in clinic-based settings. Individuals with moderate to severe hypertension face particular challenges with consistent care engagement due to the need for frequent clinic visits to titrate medications.

- We sought to determine whether community-based hypertension treatment, delivered through community health worker home visits and supervised by a clinician via telehealth, could improve hypertension control more effectively than primary care clinic-based treatment among adults with moderate–severe hypertension in rural Uganda and Kenya.

## What did the researchers do and find?

- We conducted a randomized trial in rural Kenya and Uganda among 200 adults identified to have moderate–severe hypertension through door-to-door screening by a community health worker.

- Participants were randomly allocated to receive hypertension care through either community health worker–facilitated telehealth visits in their home or traditional clinic-based care within a government-run primary care clinic in their community. All participants had free access to antihypertensive medications that were prescribed by clinicians according to country guidelines.

- We found that 77% of telehealth participants achieved hypertension control measured in all participants at 24 weeks, compared to 51% in the clinic-based group.

- At 48 weeks, 86% of participants in the telehealth group had controlled blood hypertension versus 44% in the clinic-based group, with better retention and visit adherence in the telehealth arm.

## What do these findings mean?

- In rural settings in East Africa, community health worker–facilitated telehealth improved hypertension control compared to clinic-based care among adults with moderate to severe hypertension.

- Telehealth combined with home visits by community health workers can overcome transportation barriers and improve patient engagement in the health system, leading to improved outcomes over time.

- The main limitation of our study is that it was a small pilot study with relatively few patients. This means that results might not apply to all settings or populations. To make sure our findings are as relevant as possible, we conducted the study in three different government-run clinics located in two countries.

- Further research is needed to assess cost-effectiveness and implementation at scale in routine healthcare settings.

## Introduction

Hypertension is the leading cause of cardiovascular disease (CVD) globally and is highly prevalent in Africa, with an estimated 100 million adults affected [1,2]. However, the current hypertension diagnosis, treatment, and blood pressure control remain alarmingly low, with fewer than 10% of people with hypertension achieving hypertension control [3–5]. Without effective treatment, people with severe hypertension are at especially high risk of short-term morbidity and mortality from strokes, heart disease, and kidney failure [6].

In rural Kenya and Uganda, hypertension care is predominantly available in specialty clinics within regional referral hospitals, but accessing these specialty clinics poses a significant challenge due to long travel distances and extended wait times [7]. Although multiple studies have demonstrated successful delivery of hypertension treatment in community-based primary care clinics by nurses or advanced practice providers [8–10], these clinics often face barriers such as lack of functional blood pressure cuffs and antihypertensive medications [7]. Additionally, primary care clinics in this setting typically focus on the episodic care for acute conditions and lack the infrastructure needed for robust, ongoing management of chronic conditions such as hypertension. A promising approach for strengthening hypertension care in these settings involves adapting the longitudinal care systems originally developed for chronic HIV treatment. This approach not only aims to improve CVD prevention among people with HIV, who are at a higher risk for CVD, but also applies the successful elements of the HIV chronic care model to hypertension treatment for all adults, irrespective of their HIV status [11,12]. We have previously shown that population-level hypertension screening combined with integrated care within primary care

can improve hypertension control and reduce mortality in resource-limited settings such as rural Kenya and Uganda, with the most substantial mortality benefits observed in individuals with severe baseline hypertension [13].

Although delivering hypertension treatment within primary care clinics can bring care closer to where people live and improve hypertension treatment outcomes, several barriers to accessing clinic-based longitudinal hypertension care persist. These include transportation challenges, competing priorities, medication availability and costs, and knowledge about hypertension [7]. These barriers are compounded when hypertension is severe and, thus, may require frequent visits to titrate medications. Community-based hypertension care outside of a clinic setting could address many barriers, facilitating ongoing care engagement. Prior studies in Africa have shown that community health workers (CHWs) can be trained to correctly measure blood pressure in the community and can improve hypertension screening, linkage to clinic-based care, and medication adherence [14–20]. Two large studies in lower-middle-income countries in South Asia (India, Bangladesh, Pakistan, Sri Lanka) demonstrated substantial reductions in blood pressure and improvements in hypertension control with CHW home visits to provide hypertension screening, behavioral counseling, and adherence support alongside robust referrals to clinic-based treatment [21,22]. Another study in India found that while CHW home visits for adherence support improved medication adherence, counseling alone without medication delivery was not enough to improve hypertension control [23].

Other studies have evaluated the expansion of hypertension treatment into the community, without requiring clinic attendance. A study in Uganda showed that in-person visits by a clinician and a CHW in the community improved access to care and reduced blood pressure more effectively than district hospital-based care [24]. Another study in Colombia and Malaysia showed that a CHW-led, primary care physician-supervised intervention to screen for CVD risk and initiate treatment in the community lowered 10-year CVD risk and improved blood pressure control compared to clinic-based care [25]. We aimed to build on this prior work to determine whether community-based hypertension treatment, delivered through CHW home visits and supervised by a clinician via telehealth, could improve hypertension control more effectively than primary care clinic-based treatment among adults with moderate–severe hypertension in rural Uganda and Kenya.

## Methods

### Study design and participants

Between May 12, 2022, and October 16, 2023, we conducted an individual randomized trial testing a clinician-driven, CHW-facilitated telehealth intervention for severe hypertension management compared to primary care, clinic-based hypertension care in rural Kenya and Uganda. We obtained ethical approval from the Makerere University School of Medicine Research and Ethics Committee in Uganda (2020-029); the Kenya Medical Research Institute in Kenya (P00151/4173); and the University of California San Francisco in the United States of America (20-32144). This pilot study was conducted as one of several pilot randomized trials of the SEARCH Sapphire Phase A study, registered on ClinicalTrials.gov (NCT04810650). A summary of the study protocol can be found in S2 File and the full study protocol for the parent SEARCH study can be found in S5 File.

The study took place in two communities in rural southwestern Uganda and one community in rural western Kenya. Each community had a government-run level 4 health center, which was the closest facility where community members could access primary health services. Level 4 health facilities typically provide primary care and basic inpatient and surgical services and are staffed by one physician and several clinical officers (advanced practice providers) and nurses.

Between May 12 and November 1, 2022, the local Ministry of Health (MoH) CHWs conducted door-to-door household-based hypertension screening of all adults aged ≥40 years in the community. CHWs received hands-on training in standardized blood pressure measurement with practical assessment of competencies by study clinicians prior to study start. During household visits, CHWs measured the blood pressure using automated sphygmomanometers. All adults aged ≥40 years received a single blood pressure measurement taken after at least 5 minutes of rest; those with

blood pressure ≥140 mmHg systolic or ≥90 mmHg diastolic had two additional measurements taken separated by at least 1 minute of rest [26]. Consistent with guidelines and to enhance the feasibility of large-scale screening, participants with a normal blood pressure on the first reading did not require subsequent readings [26]. Everyone with an elevated blood pressure was referred to the nearest government primary health center; those with blood pressure ≥160/100 mmHg were given a transportation voucher redeemable for approximately $5 USD upon linkage to hypertension care, based on our prior work demonstrating enhanced linkage with the use of transport vouchers [16].

Upon linkage to the health facility, study clinicians repeated blood pressure measurements using the same procedures. Participants were eligible for the study if blood pressure was elevated at both community-based and clinic measurement (≥140 mmHg systolic or ≥90 mmHg diastolic) and moderate to severely elevated on at least one of these measurements (≥160 mmHg systolic or ≥100 mmHg diastolic). Persons who were pregnant were excluded from study eligibility and were, instead, immediately linked to care within the prenatal care clinic. Participants with the evidence of possible hypertensive urgency/emergency (BP ≥ 180/110 mmHg with possibly related symptoms, or BP ≥ 200/120 mmHg) received immediate evaluation and treatment in accordance with the country guidelines within the MoH clinic prior to the initiation of any study procedures. Participants completed the written informed consent prior to enrollment.

### Randomization and masking

After receiving treatment at an initial clinic visit for hypertension care, participants were randomly assigned to receive follow-up hypertension care via telehealth (intervention condition) or in the clinic (comparison condition) using sequentially numbered sealed envelopes that when opened by the participant revealed the trial arm. The randomization sequence was generated by an independent statistician and was stratified on site and sex and implemented with a stratified block design with random block sizes of 2 and 4. Participants, clinicians, and research assistants conducting endpoint assessments were not blinded to the randomization arm, but the study statistician (LBB) was blinded until trial completion and analysis.

### Procedures

**Study enrollment and hypertension care delivery.**  All participants completed an initial in-person clinic visit for hypertension evaluation and treatment by a study clinician (a Clinical Officer) stationed at the government primary health facility in their community. At the enrollment visit, clinical evaluation included symptom and physical exam screening for cardiovascular disease complications, blood pressure measurement, height and weight measurement, and laboratory assessment that included serum creatinine, random blood glucose, lipids, urine dipstick, and offer of HIV testing (if not already known to be living with HIV). The estimated glomerular filtration rate was calculated at baseline using the 2021 CKD-EPI Creatinine equation [27].

Hypertension treatment at the initial visit and all follow-up visits was provided using a hypertension treatment algorithm developed based on country guidelines using locally available antihypertensive medications (S2 File). In primary health centers in Kenya and Uganda, hypertension medications are typically available for a small fee (~$1 US dollar per 1-month supply). For the study, all medications were provided free of charge to study participants in both the telehealth and clinic arms. In the event of medication stock-outs, the study provided additional medication supply to ensure uninterrupted access to medications. In both telehealth and clinic-based arms, follow-up visits were scheduled every 4 weeks if hypertension was uncontrolled and every 12 weeks if hypertension was controlled (defined as blood pressure <140/90 mmHg, based on country guidelines) [28,29]. Visits could be made by telehealth or in-person for the telehealth arm, with telehealth being the default unless the clinician determines that an in-person visit is needed. All clinic arm visits were scheduled in the clinic.

**Telehealth intervention arm.**  Telehealth intervention participants received follow-up care via a telehealth visit with a study clinician based at their local MoH primary health center. Telehealth visits were facilitated by a MoH CHW who

measured the participant's blood pressure, conducted an adherence assessment including pill count, provided supportive medication adherence counseling (e.g., assessing and addressing medication storage, strategies to remember pill-taking), and called the clinician for telephone consultation. Clinician involvement included the review of the information provided by the CHW, speaking with the participant to assess symptoms and provide counseling, and prescribing medications. The CHW then provided the participant with pre-packaged medications at the direction of the clinician. Prior to the visit, clinicians packed medications on a weekly basis in anticipation of potential medication combinations that might be needed by upcoming participants, based on anticipated hypertension treatment algorithm steps depending on whether their blood pressure was controlled or not (e.g., 90-day supply of current medications packaged in the event a participant had controlled blood pressure and a 30-day supply of the next algorithm step in the event blood pressure was uncontrolled and the clinician decided to intensify the treatment, with unused medications returned to the clinic after the visit). The clinician also had the option of asking the patient to come to the clinic for an in-person visit if clinically indicated. Diabetes care was also offered through telehealth, with CHWs measuring random blood glucose using portable glucometers and medication delivery based on the clinician's prescription (clinical treatment guidelines in S3 File). We were not able to deliver HIV care through the telehealth model, so participants with co-occurring HIV continued to receive HIV care at the local MoH primary health center or other clinic of their choice. CHWs recorded information on clinical hypertension visits on smartphone-based hypertension telehealth visit forms programmed using Open Data Kit [30]. Clinicians also recorded clinical hypertension data from the telehealth evaluation on tablet-based case report forms and in the paper-based MoH clinic medical record. CHWs were provided with a smartphone to facilitate telehealth calls and received a stipend to compensate for their time.

**Clinic-based comparator arm.** Participants in the clinic-based arm received follow-up hypertension care at the government MoH-run primary health facility by a study clinician. In rural health facilities in Kenya and Uganda, HIV care is imbedded within primary health centers; thus, participants with co-occurring HIV were able to receive HIV care from the same clinic. For study participants, clinic-based care was provided according to the principles of patient-centered care that we have previously described [13]. Patient-centered hypertension care included 1) nursing triage to expedite visits, 2) integrated care together in the same visit with HIV and diabetes care, as relevant, and 3) case-based provider training on creating a friendly, welcoming environment.

**Implementation strategies.** The telehealth intervention included multiple implementation strategies on top of shared strategies for both arms. Telehealth strategies included CHW home visits for blood pressure measurement and medication delivery based on clinician prescription, clinician telehealth consultation that was facilitated by the CHW, CHW training, and CHW stipend payment to support time delivering the intervention. Shared strategies in both telehealth and clinic-based arms included baseline CHW home-based hypertension screening of all adults in the community, transport reimbursement to link to the clinic for the initial visit, free hypertension medications, implementation of a country guideline-based standardized hypertension treatment algorithm, use of an electronic health record for hypertension visits, and case-based provider training on universal positive regard and friendly services. We provide a detailed list of strategies in Table A in S1 File.

**Measurements.** We conducted study visits at 24 and 48 weeks of follow-up in all participants, regardless of engagement in clinical hypertension care. These study visits were preferentially conducted in the participant's home, although they could be conducted in the clinic if participants presented to the clinic during the study visit window (±2 weeks). Study visits were conducted by research assistants who were uninvolved in the delivery of hypertension care and included blood pressure measurement using above-described measurement protocol, sociodemographic surveys, and open-ended assessment of barriers and facilitators with coded responses to capture responses. Research assistants made three call/text attempts on different days to schedule study visits at the participant's home; if unsuccessful, up to three in-person attempts were made to locate participants for weeks 24 and 48 study visits. Week 24 endpoint visits took place from October 27, 2022, to April 21, 2023. Week 48 endpoint visits took place between April 13 and October 16, 2023.

## Outcomes

The primary study outcome was blood pressure control at week 24. Study staff measured blood pressure using the same standardized procedures described above. Control was defined as blood pressure <140 mmHg systolic and <90 mmHg diastolic, consistent with country guidelines in Kenya and Uganda [28,29]. In the primary analysis, participants without week 24 blood pressure measures were assumed to be uncontrolled. Secondary outcomes included hypertension control at 48 weeks, moderate-to-severe hypertension (average BP ≥ 160/100 mmHg) at 24 and 48 weeks, retention in care at 24 and 48 weeks, and mean systolic blood pressure at 24 and 48 weeks. Retention in care was defined as not late for the most recent scheduled hypertension clinical visit (either telehealth or in-clinic) by 30 days or more. To assess the potential intervention mechanisms, we also assessed patient-reported barriers and facilitators to hypertension care using a structured survey at 24 and 48 weeks. We also report the number of attended visits for hypertension care (in-clinic or telehealth), proportion of 'time in care' defined as the proportion of follow-up time adherent to visit schedules [31], and patient-reported medication adherence over the prior 7 days across clinical visits as assessed by CHWs (telehealth arm) or clinicians (clinic-based arm). Because our trial was focused on two different strategies for delivering standard-of-care hypertension treatment, adverse event monitoring was limited to hospitalizations and deaths.

## Statistical analysis

Sample size and power calculations were based on standard formulas for a two-sample test of proportions and done with power.prop.test in R [32]. We estimated that 200 participants (~100/arm) would provide 80% power to detect at least a 20% absolute increase in hypertension control from 40% under the standard-of-care.

We assessed the intervention effect using targeted minimum loss-based estimation (TMLE) [33], which provides precision and power gains over an unadjusted effect estimator [34–37]. Within TMLE, we used Adaptive Pre-specification to select the optimal adjustment approach via sample-splitting; candidate adjustment variables were age, sex, baseline hypertension severity, and country. For all endpoints, we tested the null hypothesis of no change in outcomes using a two-sided test at the 5% significance level. For all endpoints, we examined the intervention effect in subgroups of sex, age (<60 versus ≥60 years), country, baseline hypertension severity, hypertension history (new versus prior diagnosis), baseline use of medication for hypertension, and presence of co-morbidities (HIV, diabetes, chronic kidney disease). In sensitivity analyses, we varied our approach to missing data and covariate adjustment; further details are found in the Statistical Analysis Plan (S4 File).

## Results

We screened 2,965 adults aged ≥40 years for hypertension through systematic household visits conducted by CHWs in three rural communities in Kenya and Uganda (Fig 1). One-quarter of those screened had elevated blood pressure (≥140/90 mmHg on average of three measures; 750/2965, 25%). Those with blood pressure ≥140/90 mmHg but <160/100 mmHg (n = 484) were referred to the nearest government primary health center for evaluation and treatment but were not eligible for the study. Nine percent of those screened in the community (n = 266/2965) had moderate–severe hypertension (average blood pressure ≥160/100 mmHg) and were referred to the nearest government primary health center for both clinical evaluation and screening for study enrollment. Of those with moderate–severe hypertension on initial screening, 214 (80%) linked to clinic, and 200 were enrolled in the study (98 telehealth arm, 102 clinic-based arm). Characteristics of participants who did and did not link are shown in Table B in S1 File.

The mean blood pressure of enrolled participants was 167/100 mmHg (standard deviation [SD] 17/11) at community-based screening and 161/96 mmHg (SD 20/11) on repeat measurement in the clinic (Table 1). At enrollment, 36 participants (18%) had evidence of hypertensive urgency/emergency (BP ≥ 180/110 mmHg with possibly related symptoms, or BP ≥ 200/120 mmHg) requiring immediate treatment prior to enrollment. The median age was 62 years (Q1–Q3 51–72), and most enrolled participants were women (70%; n = 139/200). Fewer than half had previously been diagnosed with hypertension (43%; n = 86/200), and only 20% (n = 39/200) reported taking antihypertensive medications in

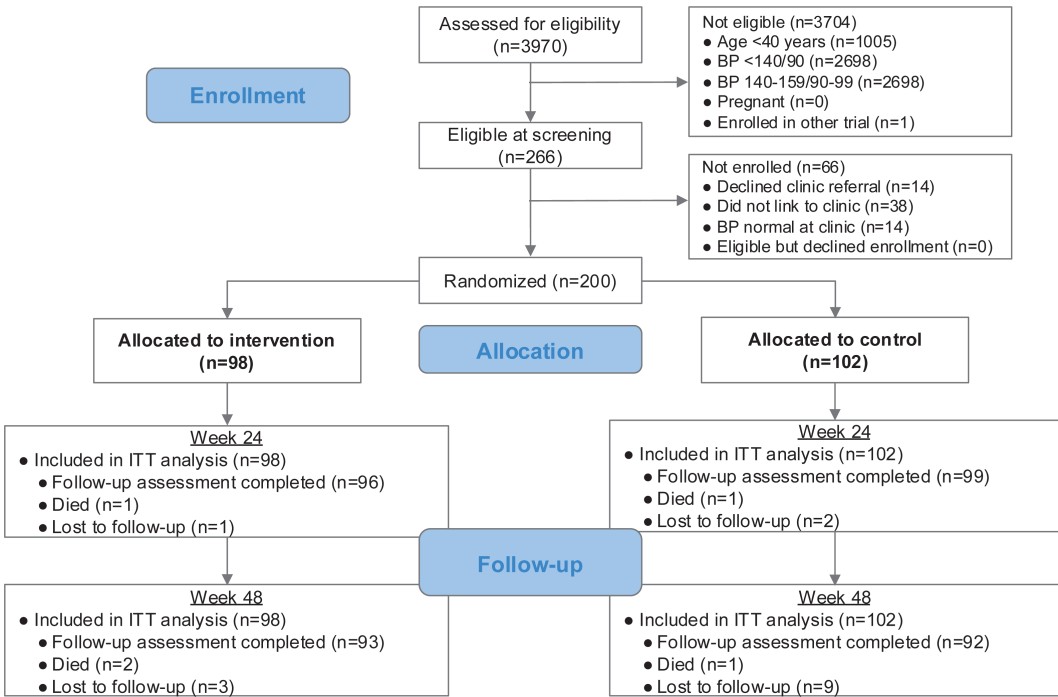

**Fig 1. Consort diagram.** All randomized participants were included in primary intention-to-treat (ITT) analysis. For hypertension control outcomes, participants with missing blood pressure measurements were assumed to have uncontrolled hypertension. We conducted additional sensitivity analyses to evaluate robustness of findings to this assumption. Abbreviations: ITT, intention-to-treat; BP, blood pressure.

the 7 days prior to enrollment. Baseline co-morbidities identified by either self-report or laboratory testing included diabetes (n = 19/200, 10%), HIV (n = 27/174, 15.5%), and chronic kidney disease (n = 45/200, 23%). No participants reported a known history of cardiovascular disease, including stroke, heart attack, or heart failure. Baseline covariates were balanced by the arm with the exception of hypertension stage; by chance, a higher proportion of telehealth intervention participants had Grade 1 hypertension (50%; n = 49/98) than clinic-based arm participants (37.3%; n = 38/102; Table 1).

HIV testing was offered at the initial clinic visit as part of integrated service delivery; 100 (50%) underwent testing, 26 (13%) were not tested due to self-reported known HIV-positive status, and 74 declined HIV testing (37%; n = 26 declined testing with self-reported negative test in prior 6 months, n = 22 declined testing with self-reported negative test >6 months, n = 26 declined testing and self-reported no history of prior testing). One person was newly diagnosed with HIV at study enrollment and was linked to same day HIV treatment. Among the 26 participants with known HIV at baseline, all reported being in care and 23 of 26 with available viral load data were virally suppressed to <200 copies/mL.

### Primary and secondary outcomes

We conducted week 24 study assessments in 195/200 (97%) of both telehealth and clinic arm participants (Fig 1). Hypertension control at 24 weeks (<140/90 mmHg; primary outcome) was 77% in the telehealth arm and 51% in the clinic-based arm (risk difference (RD) 26%, 95% confidence interval (CI) [14%, 38%], p < 0.001; Table 2). Results were robust to handling of missing data and approach to covariate adjustment (Tables C-D in S1 File). The prevalence of moderate–severe hypertension (≥160/100 mmHg) at 24 weeks was 7% in the telehealth arm and 25% in the clinic arm (RD −17%, 95% CI [−27%, −8%], p < 0.001). Retention in care at 24 weeks was 90% in the telehealth and 60% in the clinic arm (RD 30%, 95% CI [19%, 41%], p < 0.001).

**Table 1. Baseline Characteristics by Randomization Arm.**

| | Number of participants (%) | | |
|---|---|---|---|
| | Intervention (n = 98) | Control (n = 102) | Total (n = 200) |
| Country | | | |
| Kenya | 50 (51.0%) | 50 (49.0%) | 100 (50.0%) |
| Uganda | 48 (49.0%) | 52 (51.0%) | 100 (50.0%) |
| Age (median, Q1–Q3) | 60 (50–70) | 63 (53–73) | 62 (51–72) |
| Sex | | | |
| Female | 69 (70.4%) | 70 (68.6%) | 139 (69.5%) |
| Male | 29 (29.6%) | 32 (31.4%) | 61 (30.5%) |
| Body Mass Index | | | |
| Underweight (BMI < 18.5) | 13 (13.3%) | 12 (11.8%) | 25 (12.5%) |
| Normal (BMI 18.5–24.9) | 58 (59.2%) | 54 (52.9%) | 112 (56.0%) |
| Overweight (BMI 25–29.9) | 17 (17.3%) | 27 (26.5%) | 44 (22.0%) |
| Obese (BMI ≥30) | 10 (10.2%) | 9 (8.8%) | 19 (9.5%) |
| Blood pressure ever measured prior to screening | 65 (66.3%) | 63 (61.8%) | 128 (64.0%) |
| Self-reported prior hypertension diagnosis | 43 (43.9%) | 43 (42.2%) | 86 (43.0%) |
| Self-reported taking hypertension medications at screening | 16 (16.3%) | 23 (22.5%) | 39 (19.5%) |
| Diabetes mellitus | 8 (8.2%) | 7 (6.9%) | 15 (7.5%) |
| Self-reported diagnosis | 6 (6.1%) | 5 (4.9%) | 11 (5.5%) |
| Random glucose ≥11 mmol/L | 4 (4.1%) | 3 (2.9%) | 11 (5.5%) |
| HIV | 12/85 (14.1%) | 15/89 (16.9%) | 27/174 (15.5%) |
| Cardiovascular disease (self-reported history of heart attack/stroke) | 0 (0%) | 0 (0%) | 0 (0%) |
| Chronic kidney disease* | | | |
| Any CKD (stage 3+ or urine protein 1+ or greater) | 22 (22.4%) | 23 (22.5%) | 45 (22.5%) |
| Proteinuria (≥1+) | 14 (14.3%) | 14 (13.7%) | 28 (14.0%) |
| Stage 3a | 4 (4.1%) | 8 (7.8%) | 12 (6.0%) |
| Stage 3b | 5 (5.1% | 1 (1.0%) | 6 (3%) |
| Stage 4 | 0 (0%) | 0 (0%) | 0 (0%) |
| Stage 5 | 1 (1.0%) | 2 (2.0%) | 3 (1.5%) |
| Blood pressure at community-based screening (mmHg) | | | |
| SBP (screening) | | | |
| Mean (SD): | 166 (17) | 168 (17) | 167 (17) |
| Median (Q1–Q3): | 163 (154–176) | 164 (160–173) | 164 (159–174) |
| DBP (screening) | | | |
| Mean (SD): | 101 (10) | 99 (12) | 100 (11) |
| Median (Q1–Q3): | 102 (94–107) | 100 (92–108) | 101 (93–107) |
| Blood pressure at clinic enrollment (mmHg) | | | |
| SBP (enrollment) | | | |
| Mean (SD): | 160 (20) | 162 (19) | 161 (19) |
| Median (Q1–Q3): | 152 (146–172) | 160 (149–174) | 157 (147–173) |
| DBP (enrollment) | | | |
| Mean (SD): | 96 (10) | 96 (13) | 96 (11) |
| Median (Q1–Q3): | 95 (89–102) | 96 (89–103) | 96 (89–102) |
| Hypertension stage at enrollment† | | | |
| 140–159/90–99 mmHg | 49 (50.0%) | 38 (37.3%) | 87 (43.5%) |
| 160–179/100–109 mmHg | 23 (23.5%) | 40 (39.2%) | 63 (31.5%) |

*(Continued)*

**Table 1.** (Continued)

| | Number of participants (%) | | |
| --- | --- | --- | --- |
| | Intervention (n = 98) | Control (n = 102) | Total (n = 200) |
| ≥180/110 mmHg | 26 (26.5%) | 24 (23.5%) | 50 (25.0%) |
| -Asymptomatic | 11 (11.2%) | 8 (7.8%) | 19 (9.5%) |
| -Symptomatic‡ | 15 (15.3%) | 16 (15.7%) | 31 (15.5%) |

Abbreviations: Q1, first quartile; Q3, third quartile; BMI, body mass index; HIV, human immunodeficiency virus; CKD, chronic kidney disease; SBP, systolic blood pressure; DBP, diastolic blood pressure; SD, standard deviation; mmHg, millimeters of mercury.

Note: The denominator for all variables in Table 1 is the same as the column total, except for HIV testing where we excluded those who had an unknown HIV status and declined testing.

*CKD stage was determined using glomerular filtration rate (GFR) categories defined by the Kidney Disease Improving Global Outcomes (KDIGO) CKD Work Group [38].

†Participants were eligible for the study if blood pressure was ≥ 160/100 mmHg at community screening and remained ≥140/90 mmHg at initial clinic visit, or if blood pressure was ≥ 160/100 mmHg at initial clinic visit.

‡Symptoms included: severe headache (n = 20), vision changes (n = 16), chest pain (n = 9), shortness of breath (n = 3), leg swelling (n = 6), recent loss of consciousness (n = 1).

**Table 2. Study outcomes.**

| | Intervention proportion/mean [95% CI] | Control proportion/mean [95% CI] | Effect estimates [95% CI] |
| --- | --- | --- | --- |
| *24 weeks* | | | |
| Hypertension control (<140/90 mmHg) | 77% [69%, 85%] | 51% [42%, 60%] | 26% [14%, 38%]; p < 0.001 |
| Moderate–severe HTN (≥160/100 mmHg) | 7% [2%, 12%] | 25% [16%, 33%] | −17% [−27%, −8%]; p < 0.001 |
| Retention in care | 90% [85%, 96%] | 60% [51%, 69%] | 30% [19%, 41%]; p < 0.001 |
| Mean SBP (mmHg) | 133 [130, 137] | 141 [137, 145] | −7.5 [−12.9, −2.1]; p = 0.006 |
| *48 weeks* | | | |
| Hypertension control (<140/90 mmHg) | 86% [79%, 92%] | 44% [34%, 54%] | 42% [30%, 53%]; p < 0.001 |
| Moderate–severe HTN (≥160/100 mmHg) | 2% [0%, 4%] | 15% [8%, 22%] | −13% [−21%, −6%]; p < 0.001 |
| Retention in care | 83% [75%, 90%] | 50% [41%, 60%] | 32% [20%, 44%]; p < 0.001 |
| Mean SBP (mmHg) | 133 [130, 136] | 141 [138, 145] | −8.2 [−12.8, −3.7]; p < 0.001 |
| Time in care | 89% [86%, 92%] | 63% [56%, 69%] | 26% [19%, 35%]; p < 0.001 |

Abbreviations: HTN, hypertension; SBP, systolic blood pressure; CI, confidence interval; mmHG, millimeters of mercury.

At week 48, we completed end-of-study assessments in 93/98 telehealth participants (95%) and 92/102 clinic-based participants (90%). Hypertension control (<140/90 mmHg) was 86% in the telehealth group and 44% in the clinic group (RD 42%, 95% CI [14%, 38%], p < 0.001). The prevalence of moderate–severe hypertension (≥160/100 mmHg) at 48 weeks was 2% in the telehealth arm and 15% in the clinic arm (RD −13%, 95% CI [−21%, −6%], p < 0.001). Fig 2 shows the hypertension severity, and Fig A in S1 File shows the mean blood pressure by study arm at enrollment, week 24, and week 48. Retention in care at 48 weeks was 83% in the telehealth arm and 50% in the clinic arm (RD 32%, 95% CI [20%, 44%], p < 0.001). In *post hoc* effectiveness analyses, time in care over 48 weeks was 89% in the telehealth arm and 63% in the clinic arm (difference 26%, 95% CI [19%, 34%], p < 0.001).

Similar intervention effects on control at 24 and 48 weeks were seen across subgroups, defined by age, sex, baseline hypertension severity, hypertension history, and presence of co-morbidities (HIV, diabetes, chronic kidney disease); those with baseline hypertension medication use had a smaller telehealth intervention effect, driven by improved outcomes in

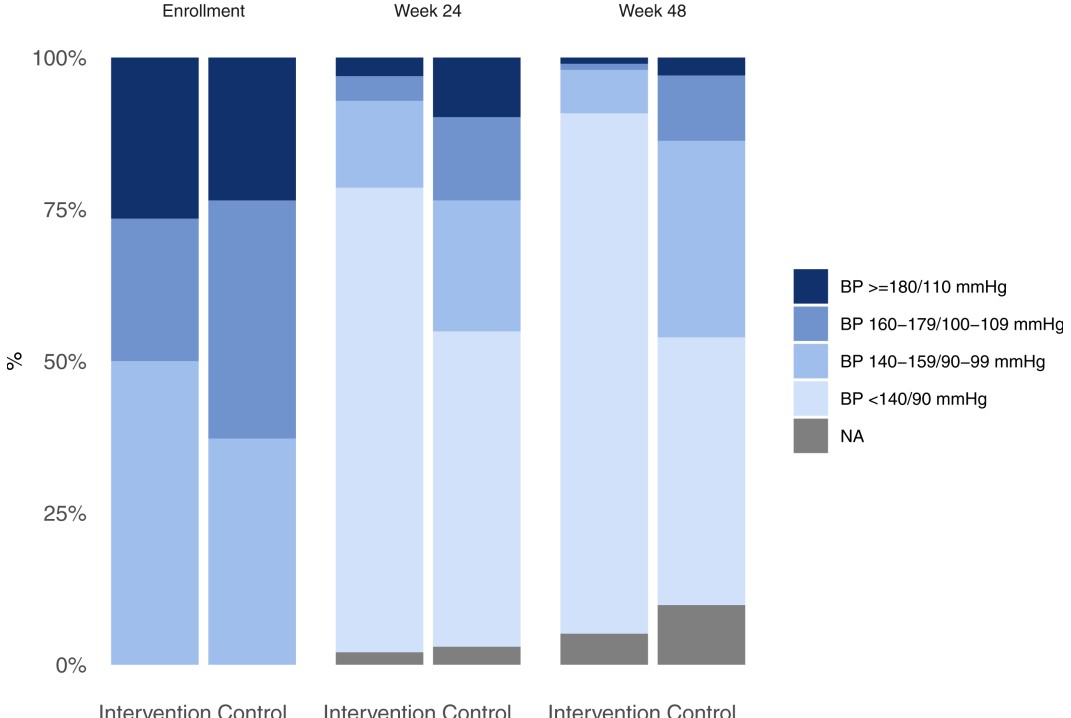

**Fig 2. Hypertension severity.** Measured among all study participants (n = 200). Week 24 and week 48 measurements were taken by research assistants who were not involved in delivering hypertension care through the telehealth intervention or clinic-based comparator condition. NA indicates missing measurements at week 24 or week 48 study visits (refer to Fig 1). Underlying data can be found in Table K in S1 File. Abbreviations: BP, blood pressure; mmHg, millimeters of mercury; NA, not applicable (missing).

the clinic-based arm for this group (Fig 3 and Figs B-C in S1 File). Although all participants with HIV enrolled were virally suppressed at baseline and well engaged in HIV care at enrollment, their retention in hypertension care at 48 weeks varied markedly by arm: 100% in the telehealth arm but only 53% in the clinic-based arm (RD 47%, 95% CI [20%, 74%], p = 0.002; Fig E in S1 File).

### Self-reported barriers and facilitators of hypertension care engagement

At 48 weeks, 69% of clinic arm participants reported transportation challenges to be a barrier to consistent engagement in hypertension care; only 2% of telehealth arm participants noted transportation as a barrier. Participants randomized to clinic-based care were more likely to report barriers related to their hypertension diagnosis than intervention participants (17% versus 6%), including not wanting to take medications due to concerns about side effects (8% versus 4%), feeling well and not wanting to take medications (12% versus 1%), or did not believe that they had hypertension (6% versus 0%). Notably, only 2% of participants identified issues with care quality as a barrier to care (same in both arms). Facilitators for care engagement were similar by trial arm except for transportation (reported as a facilitator in 90% of intervention and 5% of control participants). See Figs F-G in S1 File.

### Hypertension care delivery

Based on blood pressure measures at clinical hypertension visits (telehealth or clinic-based), analysis of the median time to hypertension control was similar in both arms (56 days in telehealth intervention arm, 57 days in clinic-based

## A. Hypertension control at week 24

| | N | Intervention | Control | Risk difference (95% CI) | Favors intervention | p-value |
|---|---|---|---|---|---|---|
| All | 200 | 77% (69,85) | 51% (42,60) | 0.26 (0.14,0.38) | | <0.01 |
| **Country** | | | | | | |
| Kenya | 100 | 70% (57,83) | 36% (23,49) | 0.34 (0.15,0.52) | | <0.01 |
| Uganda | 100 | 84% (75,94) | 66% (53,79) | 0.19 (0.03,0.35) | | 0.02 |
| **Sex** | | | | | | |
| Women | 139 | 80% (71,89) | 52% (41,63) | 0.28 (0.14,0.42) | | <0.01 |
| Men | 61 | 71% (55,86) | 48% (31,66) | 0.22 (−0.01,0.45) | | 0.06 |
| **Age Group** | | | | | | |
| Age 40–59 yr | 82 | 76% (63,88) | 42% (27,56) | 0.34 (0.15,0.53) | | <0.01 |
| Age 60+ yr | 118 | 78% (67,88) | 57% (45,70) | 0.20 (0.04,0.37) | | 0.01 |
| **Baseline hypertension severity** | | | | | | |
| Grade 1 | 87 | 82% (71,93) | 52% (36,67) | 0.31 (0.12,0.49) | | <0.01 |
| Grade 2 | 63 | 84% (72,97) | 54% (40,67) | 0.31 (0.12,0.49) | | <0.01 |
| Grade 3 | 50 | 62% (43,80) | 46% (26,66) | 0.16 (−0.12,0.43) | | 0.26 |
| **Prior diagnosis of hypertension (HTN)** | | | | | | |
| History of HTN | 86 | 74% (61,87) | 58% (44,73) | 0.16 (−0.03,0.35) | | 0.11 |
| No history of HTN | 114 | 79% (69,89) | 47% (35,59) | 0.32 (0.17,0.48) | | <0.01 |
| **Baseline hypertension medication** | | | | | | |
| On HTN medication | 39 | 75% (53,97) | 70% (50,89) | 0.05 (−0.24,0.35) | | 0.71 |
| Not on HTN medication | 161 | 77% (68,86) | 46% (36,56) | 0.31 (0.17,0.45) | | <0.01 |

0.00  0.25  0.50  0.75  1.00
Risk difference

## B. Hypertension control at week 48

| | N | Intervention | Control | Risk difference (95% CI) | Favors intervention | p-value |
|---|---|---|---|---|---|---|
| All | 200 | 86% (79,92) | 44% (34,54) | 0.42 (0.30,0.53) | | <0.01 |
| **Country** | | | | | | |
| Kenya | 100 | 85% (75,95) | 35% (24,47) | 0.49 (0.34,0.65) | | <0.01 |
| Uganda | 100 | 85% (77,94) | 56% (42,69) | 0.30 (0.14,0.46) | | <0.01 |
| **Sex** | | | | | | |
| Women | 139 | 88% (81,96) | 46% (34,57) | 0.43 (0.29,0.57) | | <0.01 |
| Men | 61 | 80% (66,95) | 40% (23,57) | 0.41 (0.18,0.63) | | <0.01 |
| **Age Group** | | | | | | |
| Age 40–59 yr | 82 | 93% (86,101) | 41% (26,56) | 0.52 (0.35,0.69) | | <0.01 |
| Age 60+ yr | 118 | 80% (70,90) | 44% (32,57) | 0.36 (0.20,0.52) | | <0.01 |
| **Baseline hypertension severity** | | | | | | |
| Grade 1 | 87 | 90% (81,98) | 53% (37,69) | 0.37 (0.19,0.55) | | <0.01 |
| Grade 2 | 63 | 96% (87,104) | 40% (25,55) | 0.56 (0.38,0.73) | | <0.01 |
| Grade 3 | 50 | 69% (51,87) | 38% (18,57) | 0.32 (0.05,0.58) | | 0.02 |
| **Prior diagnosis of hypertension (HTN)** | | | | | | |
| History of HTN | 86 | 86% (76,96) | 48% (33,63) | 0.38 (0.20,0.56) | | <0.01 |
| No history of HTN | 114 | 86% (78,95) | 39% (26,52) | 0.48 (0.32,0.63) | | <0.01 |
| **Baseline hypertension medication** | | | | | | |
| On HTN medication | 39 | 81% (61,101) | 65% (45,86) | 0.16 (−0.13,0.45) | | 0.26 |
| Not on HTN medication | 161 | 87% (80,94) | 38% (27,49) | 0.49 (0.36,0.62) | | <0.01 |

0.00  0.25  0.50  0.75  1.00
Risk difference

**Fig 3. Hypertension control by subgroup.** (A) Hypertension control at 24-week study visit, stratified by subgroup. (B) Hypertension control at 48-week study visit, stratified by subgroup. Sub-group analyses were prespecified in the statistical analysis plan, with the exception of prior diagnosis of hypertension and baseline hypertension medication which were *post hoc* analyses requested during peer review. Abbreviations: yr, years; HTN, hypertension; CI, confidence interval.

comparator arm). By 24 weeks, 91% (95% CI [83%, 95%]) of telehealth intervention participants achieved hypertension control during at least one clinical hypertension visit compared to 69% (95% CI [58%, 76%]) for clinic-based comparator arm participants (Fig D in S1 File).

Over 48 weeks of follow-up, all telehealth participants had at least one post-baseline follow-up visit for hypertension treatment, compared to 80% of clinic-based participants (80/102). Telehealth participants had a median of 6 visits (Q1–Q3 5–7, range 2–10) and clinic participants 5 visits (Q1–Q3 2–6, range 1–13), including the enrollment visit. In the telehealth group, 95% of post-baseline visits were conducted at home by telehealth (485/512), and only 5% of visits were conducted in the clinic due to clinician recommendation (27/512). All post-baseline visits in the clinic-based arm occurred in the clinic (373/373). Nearly all participants were prescribed at least one antihypertensive medication (91% intervention, 93% control), and most were prescribed ≥2 medications (60% telehealth, 69% clinic arms; Tables E-G in S1 File). Patient-reported medication adherence assessed during clinical visits was high in both arms (Tables H-I in S1 File).

At the end of the study, 88% of telehealth participants stated a preference to receive care at home (n = 76) or a combination of home/clinic (n = 10); 76% of clinic arm participants preferred home (n = 64) or a combination of home/clinic (n = 11).

### Adverse events

Three participants died, two in the telehealth arm (one with baseline stage 5 chronic kidney disease who died of renal failure and one who died of heart failure) and one in the clinic arm (death at home following acute illness). Two additional participants were hospitalized, both in the telehealth arm (stroke, inguinal hernia repair). The participants who died were receiving effective treatment for hypertension with last observed blood pressures controlled to <140/90 mmHg. All adverse events were determined to be unrelated to the study intervention (hypertension telehealth care). See Table J in S1 File.

### Discussion

In this randomized pilot study, we showed that clinician-driven, CHW-facilitated telehealth nearly doubled the proportion of people achieving hypertension control compared to clinic-based hypertension care. Nearly 90% of telehealth intervention participants achieved hypertension control after 48 weeks. Our clinic-based comparator arm consisted of a patient-centered hypertension care model that we previously showed an improved hypertension control and all-cause mortality compared to current standard hypertension care in primary health centers in Kenya and Uganda [13]. Thus, CHW-facilitated telehealth may lead to even greater improvements when compared to current standard hypertension care in rural MoH primary care clinics in Kenya and Uganda.

We postulate that primary mechanism through which the telehealth intervention led to improved outcomes was through improved retention in care. In particular, the CHW-facilitated telehealth intervention reduced transportation barriers to care engagement, resulting in a greater proportion of time patients attended follow-up visits on time and were in possession of and adherent to hypertension medications. Nearly all participants reported that transportation to clinic was a challenge to consistent care engagement, due to a combination of transportation costs, time, and mobility challenges. Additionally, the CHW-facilitated telehealth model of care resulted in fewer participants reporting concerns about taking medications for an asymptomatic condition or disbelief that they truly had hypertension, highlighting the value added by offering home visits with supportive adherence counseling by a local community member (CHW) in addition to counseling by a clinician.

Hypertension control in the clinic-based arm was comparable to hypertension control achieved in other facility-based interventions in the literature, including pharmacy-based hypertension care [39], task sharing to integrate mental health and hypertension care [40], and health insurance coverage with nurse-led hypertension care [10]. By attaining much higher levels of hypertension control, CHW-facilitated telehealth could augment these facility-based care models to improve hypertension treatment outcomes particularly for those living in rural settings.

We did not measure downstream CVD outcomes; although based on large meta-analyses of hypertension treatment trials, the degree of SBP reduction we observed would be expected to result in a 50%–60% reduction in incidence of heart

attacks and strokes, if sustained [6,41,42]. Our study provides evidence that CHWs in the rural East African context can successfully conduct population-level hypertension screening, identify those with moderate–severe hypertension with unambiguous indication for and large benefit from treatment (9% of population), and facilitate treatment delivery through clinician telehealth that leads to a very high degree of sustained hypertension control. We have previously shown that CHW hypertension screening and linkage to clinic-based care would improve CVD outcomes and be cost-effective at scale [43]. Scaling of our CHW-facilitated telehealth intervention alongside population-level CHW screening is likely to lead to even greater health benefits, although further study of cost and cost-effectiveness is needed.

For people with HIV, integration of HIV and hypertension care is an important way to improve cardiovascular disease prevention [44,45]. However, in our clinic-based arm where HIV and hypertension care were integrated, retention in hypertension care among people with HIV was only 53% at week 48, compared to 100% retention in the telehealth arm. Several factors likely explain low retention in hypertension care among people with HIV in our study. First, moderate–severe hypertension requires a different level of intensive monitoring until stabilized, often monthly visits until blood pressure is controlled. In contrast, HIV visits are often for every 3–6 months for stable patients. Second, only 44% (n = 12/27) of study participants with HIV and hypertension accessed HIV care at their local community primary care clinic (where the study was based). We did not assess reasons for receiving HIV care at a different clinic outside of the community, although reasons may include stigma, travel/mobility, or other preference-related factors. At the end of the study, 74% of people with HIV expressed a preference for receiving at least some hypertension visits at home through CHW-facilitated telehealth, highlighting the role this intervention could play in a patent-centered approach to hypertension care delivery even among people with HIV already accessing the clinic for their HIV care.

We focused on the use of telehealth to bridge primary care clinics to patients in the community with facilitation by a CHW, although our telehealth model could also be expanded to enhance management of complex cases. Currently, people with resistant hypertension or existing cardiovascular disease are generally referred to tertiary care settings which involve high out-of-pocket costs and long travel times for patients [46]. Telehealth has been identified as a strategy to support primary care clinicians by providing specialty consultation for complex hypertension cases [47]. Future evolution of our telehealth model could involve linking communities, primary care clinics, and specialists through telehealth consultation, with CHWs as key facilitators to ensure patients receive consistent monitoring and are retained in care.

Our study had several limitations. First, by chance, there was imbalance hypertension severity at baseline between arms. However, our primary analysis adjusted for baseline hypertension grade as well as other variables, and our results were robust to the analytic approach and sensitivity analyses (Tables C-D in S1 File). Second, although fully powered for our primary outcome, this was a pilot study with a relatively small sample size which limits inferences about effectiveness at scale. However, by conducting the study at three government primary care clinics that are broadly representative of primary health centers in rural Kenya and Uganda, our findings are likely generalizable to similar rural primary healthcare settings. We are currently conducting a follow-up population-level study to understand intervention effects at scale and when integrated with population-level screening for HIV and HIV risk (NCT05768763). Future studies will also need to evaluate cost-effectiveness of additional resource inputs including CHW stipends, medication costs, and clinician time to deliver telehealth visits. Third, this was an individual-level randomized trial in a controlled setting where the intervention was delivered by study clinicians. Future study is needed to understand the implementation strategies for incorporating CHW-facilitated telehealth for community-based hypertension treatment into routine care settings. Finally, intervention allocation was not blinded to staff delivering the interventions (CHWs, clinicians) or research assistants assessing outcomes, which could have introduced bias. However, our primary outcome of blood pressure control was assessed using automated blood pressure machines, minimizing the potential for bias in outcome assessment. Study clinicians delivering hypertension care could have differentially influenced quality of care by trial arm, although we saw no differences in the number of drugs prescribed and clinicians were trained to provide high-quality care to both trial arms.

In this pilot RCT, clinician-driven, CHW-facilitated telehealth substantially improved hypertension control compared to clinic-based hypertension care. Facilitated telehealth is a promising intervention for supplementing integrated clinic-based care to improve cardiovascular disease prevention for people both with and without HIV.

## Supporting information

**S1 File.** **Supplemental tables and figures.**
(PDF)

**S2 File.** **Summary of study protocol.**
(PDF)

**S3 File.** **Hypertension and diabetes treatment guidelines used by clinicians.**
(PDF)

**S4 File.** **Pre-specified statistical analysis plan.**
(PDF)

**S5 File.** **Full protocol for the parent study.**
(PDF)

**S6 File.** **Deidentified dataset.**
(CSV)

**S7 File.** **Data dictionary.**
(XLSX)

**S1 Consort Checklist.** **CONSORT 2010 checklist of information to include when reporting a randomised trial.**
(PDF)

## Acknowledgments

The SEARCH project gratefully acknowledges the Ministry of Health of Uganda and of Kenya, our research teams and administrative teams in San Francisco, Uganda, and Kenya, collaborators and advisory boards, and especially all communities and participants involved.

## Author contributions

**Conceptualization:** Matthew D. Hickey, Asiphas Owaraganise, Sabina Ogachi, Norton Sang, Erick M. Wafula, Jane Kabami, Gabriel Chamie, Elijah Kakande, Maya L. Petersen, Laura B. Balzer, Diane V. Havlir, Moses R. Kamya, James Ayieko.

**Data curation:** Erick M. Wafula, Nicole Sutter, Jennifer Temple, Laura B. Balzer.

**Formal analysis:** Matthew D. Hickey, Jennifer Temple, Laura B Balzer.

**Funding acquisition:** Matthew D. Hickey, Maya L. Petersen, Diane V. Havlir, Moses R. Kamya.

**Investigation:** Asiphas Owaraganise, Sabina Ogachi, Norton Sang, Jane Kabami, Anthony Muiru, Gabriel Chamie, Maya L. Petersen, Diane V. Havlir, Moses R. Kamya, James Ayieko.

**Methodology:** Matthew D. Hickey, Laura B. Balzer.

**Project administration:** Asiphas Owaraganise, Sabina Ogachi, Norton Sang, Erick M. Wafula, Jane Kabami, Nicole Sutter, James Ayieko.

**Supervision:** Jane Kabami, Anthony Muiru, Elijah Kakande, Maya L. Petersen, Diane V. Havlir, Moses R. Kamya, James Ayieko.

**Writing – original draft:** Matthew D. Hickey.

**Writing – review & editing:** Asiphas Owaraganise, Sabina Ogachi, Norton Sang, Erick M. Wafula, Jane Kabami, Nicole Sutter, Jennifer Temple, Anthony Muiru, Gabriel Chamie, Elijah Kakande, Maya L. Petersen, Laura B. Balzer, Diane V. Havlir, Moses R. Kamya, James Ayieko.

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
