## [Editor Report · Decision Letter 0]

Dear Dr Hickey,

Thank you for submitting your manuscript entitled "Randomized trial of Community Health Worker Facilitated Telehealth for Moderate-Severe Hypertension Care in Kenya and Uganda" for consideration by PLOS Medicine.

Your manuscript has now been evaluated by the PLOS Medicine editorial staff and I am writing to let you know that we would like to send your submission out for external peer review.

We kindly ask that you provide a copy of the original trial protocol and a completed CONSORT checklist as a supporting information files. By protocol, we mean the complete and detailed plan for the conduct and analysis of the trial that the ethics committee approved before the trial began. Please send this in the original language. If this is in a language other than English, please also provide a translation. Please detail any deviations from this study protocol in the Methods section of your manuscript. The documents will be made available to the editors and reviewers.

Please re-submit your manuscript within two working days, i.e. by Nov 01 2024.

Feel free to email me at atosun@plos.org or us at plosmedicine@plos.org if you have any queries relating to your submission.

Kind regards,

Alexandra Tosun, PhD

Associate Editor

PLOS Medicine

---

## [Decision Letter · Decision Letter 1]

Dear Dr Hickey,

Many thanks for submitting your manuscript "Randomized trial of Community Health Worker Facilitated Telehealth for Moderate-Severe Hypertension Care in Kenya and Uganda" (PMEDICINE-D-24-03678R1) to PLOS Medicine. The paper has been reviewed by subject experts and a statistician; their comments are included below and can also be accessed here: [LINK]

As you will see, the reviewers found the manuscript interesting and relevant, but raised a number of methodological issues and points for clarification. After discussing the paper with the editorial team and an academic editor with relevant expertise, I'm pleased to invite you to revise the paper in response to the reviewers' comments. We plan to send the revised paper to some or all of the original reviewers, and we cannot provide any guarantees at this stage regarding publication.

We ask that you submit your revision by Jan 06 2025. However, if this deadline is not feasible (including due to the upcoming holidays), please contact me by email and we can discuss a suitable alternative. Please note that the editorial team will be out of office from 23 December 2024 up to and including 3 January 2025.

Don't hesitate to contact me directly with any questions (atosun@plos.org).

Best regards,

Alexandra

Alexandra Tosun, PhD

Associate Editor

PLOS Medicine

atosun@plos.org

Comments from the academic editor:

Overall, the trial seems quite well done. However, there are some aspects that seem a little unclear.

It is not 100% clear why the trial is described as a pilot trial when it is fully powered for blood pressure outcomes. How do the authors think about why this is a pilot trial and what is the scope of the subsequent, presumably larger, trial?

One of the biggest areas of lack of clarity is the precise specification of what constitutes the intervention and what constitutes the control arm. A table or figure might help here. It would be useful to know how people in both arms of the trial were treated - whether they were treated as part of the routine government health care system, or whether they were treated by trial clinicians funded by the project, and how often they were seen. It would also be helpful to know whether the medications were paid for by the patients, the health system or the research team, and whether this differed between arms. One overarching request is that the authors should be much more specific about the implementation strategy they believe they are using in this trial. It seems that there is much more than just a telehealth intervention; there is medication delivery, home care, task sharing with community health workers, and a number of other elements that together could form a package that includes telehealth. More contextual information about the health system would be helpful. But as written, it is a bit difficult for the reader to understand exactly what was done and who paid for it. In implementation science jargon, the implementation strategies are underspecified (Proctor EK, et al. Implement Sci 2013;8:139).

There are also some important missing reporting items: medication adherence or pill count, intensity of intervention vs. control in terms of visits or some indication of how many contacts patients had, and exact medication usage in each arm, including average dose. Was adherence assessed? It would be good to see the mean blood pressure in each arm reported in Table 1 (perhaps some of this data was not collected).

One striking aspect missing from the discussion is any comment or speculation on the mechanism for the trial's putative success. In particular, the top-line results show a very large effect size, but the number of drug prescriptions was essentially the same in each arm. So the mechanism for this extremely large effect size is confusing. Was it a result of access to medication after prescription, or was it an adherence problem where patients who had access to medication did not take it? It seems unlikely that lifestyle counselling alone could account for such a dramatic difference in blood pressure control.

Authors should make sure that everything reported in the abstract is also reported in the results of the manuscript. For example, the number of antihypertensive drugs prescribed at 48 weeks is reported in the abstract, but not in the manuscript.

It is very difficult to follow the protocol and the clinicaltrials.gov page because this is one of many trials that are all registered and protocolised together. It seems that it would be very useful for readers if the protocol could be consolidated to be for this trial only. Clarity and readability is really important and this would make it easier.

Comments from the editorial team:

In line with the comments of the academic editor and the reviewers, the editorial team agrees that there is room for more clarity and detail throughout the manuscript. We also agree that, in addition to providing the full original trial protocol, it would be helpful to provide an additional supplementary file that summarises the protocol information specifically for this trial.

Comments from the reviewers:

Reviewer #1: Thank you for the opportunity to review this randomized clinical trial evaluating the effects of a HCW facilitated and clinician led telehealth intervention on blood pressure control in rural Kenya and Uganda. The report was interesting, and I appreciated the inclusion of patient feedback through end line qualitative surveys. I had some comments in reading, which are itemized below.

Introduction

1. Please add context about how hypertension is treated in this setting. 'Clinic' covers a wide variety of healthcare settings - in this context, are these community-based primary health care centers, regional clinics, or tertiary care centers? How are the facilities staffed, and how would a patient with hypertension typically be diagnosed and treated? Are primary care physicians always on site at the clinic?

2. There are others studies in low-income settings which have evaluated CHW-led hypertension care within a home/community setting. The closing arguments in this section could be expanded to include other examples of the positive effects of CHW-led hypertension care - see Joshi et al, 2019, American Heart Journal; Jafar et al, 2020 The New England Journal of Medicine; Khetan et al, 2019 Global Heart journal; O'Neil et al, 2016, Health Policy and Planning.

Methods

3. Can you clarify the exclusion of young adults in this population? i.e. aged 18-39.

4. Were medications always available? Were they subsidized in any way for participants?

5. "Participants without week 24 blood pressure measures were assumed to be uncontrolled."- have you performed a sensitivity analysis here to evaluate the impact of this assumption?

Results

6. "and were referred to the nearest government primary health center for evaluation and treatment. Nine percent of those screened (n=266) had moderate-severe hypertension (average blood pressure ≥160/100 mmHg) and were referred to the nearest government primary health center for clinical evaluation and study enrollment." I think in the second sentence here, you mean that they were referred for study enrollment, assuming they were already at the PHC based on their prior referral for evaluation?

7. Of the 750 patients who screened for hypertension in the community, how many presented at the PHC for further evaluation? There seems to be relatively high attrition from the 266 who screened for moderate-severe hypertension and the 214 who linked to clinic - was similar attrition observed at the earlier step?

8. "Of those with severe hypertension on initial screening, 214 (80%) linked to clinic and 200 were enrolled in the study (98 intervention, 102 control)." For consistency, I think this should be "moderate-severe hypertension".

9. "Mean blood pressure of enrolled participants was 167/100 mmHg at community-based screening and 161/96 mmHg on repeat measurement in the clinic." - these measures need an indicator of variability, i.e. SD.

10. Reasons for declining HIV testing - with numbers in each group - are reported twice. One of these segments should be deleted.

11. Please clarify in the adverse events that one patient died to due renal failure and one due to heart failure - as written, this is ambiguous.

Table 1

12. Where '0' is reported as the frequency, a decimal should not be used in the percent, i.e. report as 0 (0%).

13. CKD - the numbers in these subcategories do not sum to the top line of 'Any CKD (stage 3+ or urine protein 1+ or greater)'. I understand they may not be mutually exclusive, but the numbers in the control group, for example, sum to 18 (14+2+1+1) which is less than the top line reporting 22. Please clarify with footnotes or correct the numbers.

Reviewer #2: This pilot RCT sought to test whether a clinician-driven, CHW facilitated telehealth intervention would improve hypertension control among adults with severe hypertension in rural settings.

This is a relevant strategy under study, with the potential to inform strategies to improve access to chronic disease care in areas with most need.

Here are a few observations:

All participants with an elevated Bp during household screening were referred to the nearest primary health center. What was the linkage rate?

The enrollment clinical and laboratory evaluation was thorough. It is not clear if the management was based on the evaluation findings, eg, management of dyslipidemia, diabetes, CKD, etc among the study participants, given that the focus was on hypertension, yet ethically bound to manage diagnosed co-morbidities. For the participants in the intervention arm with comorbidities, what was the medicine delivery option for non-hypertension diagnoses?

What was the definition of linkage for the intervention arm?

It is also not clear how the medication dose adjustment was done for the participants in the intervention arm given that the prepacked medication was delivered by the CHW, yet a dosing decision should be based on the Bp at the household encounter.

There is an important sub-group of enrollees; those who had been previously diagnosed with hypertension (42%) and the 19% who were on medication. How did their outcomes compare with those diagnosed for the first time through the project? It would also be very informative to analyze the outcomes against the comorbidities.

Amon the HIV clients, clarify if the HIV care remained clinic based even for those in the intervention arm. In the control arm, how can the retention be 53% when they are already coming to the clinics for HIV care, unless the clinics are independent of each other, thus negating the service integration referred to.

It would be great to have clarity about the payment for medication and the overall supply chain for the intervention. Whether the project meeting the cost of drugs, or the patients were paying, and what happens when the clients had no money, and the role of health insurance if any.

The authors can comment on the utility of a single Bp reading to determine control.

In general This is a well-designed study that generates important information that can inform NCD interventions.

Reviewer #3: Overall, I found the methodology and reporting clear and robust. It is great to see high retention and follow-up rates at week 24 and week 48. My main concern relates to baseline imbalances in blood pressure and to what extent this was appropriately accounted for in the analysis.

Major/general comments:

1. According to table 1, it looks as if a greater proportion of patients had hypertension falling in the "160-179/100-109 mmHg" category in the control arm (39.2% in control vs 23.5% in intervention). This is a substantial baseline difference. While it may be due to chance, this important imbalance in the outcome of interest has the potential to bias the main outcome analysis in favor of the intervention if not appropriately adjusted for. I note that baseline hypertension grade was a candidate variable in the TLME approach; however, I would be keen to understand whether baseline hypertension grade was adjusted for in the final model and if the results change with and without adjustment. I would also strongly recommend adjusting for baseline blood pressure as a continuous variable in every analysis (forcing the inclusion as a covariate and not letting the TLME model make the selection). As it stands it is not clear to me whether the difference in hypertension control at week 24 and week 48 are simply a "carry-over" from the differences already present at baseline or a true effect of the intervention.

2. Please provide further information around the approach used to derive absolute differences and 95% confidence intervals. Given TLME was used, I expect these to be adjusted for baseline covariates rather than being raw differences in percentages. However, the modelling approach is not entirely clear to me and it currently looks as if all differences are raw (unadjusted) differences.

Minor/specific comments:

1. Please clarify whether the telehealth intervention was in addition to usual clinical care or used as a replacement? In other words, were participants randomised to the intervention still able to be followed up in the clinic? If so, please consider clarifying this in the manuscript. Please also consider renaming "control" as "usual care" or "clinical care".

2. I note that participants and clinicians were not blinded to the group allocation. Please clarify whether assessors were blinded and, if not, whether this may have caused any bias.

3. On page 6, it is written that "Regardless of the trial arm, follow-up visits were scheduled every 4 weeks if hypertension was uncontrolled and every 12 weeks if hypertension was controlled". Please clarify whether these follow-up visits were expected to be in-person clinic visits for all participants? Or was it telehealth visits for "intervention" participants and clinic visits for "control" participants?

4. In the outcomes section, it is written that "Participants without week 24 blood pressure measures were assumed to be uncontrolled". This is a fairly strong/conservative assumption about the missing data and I wonder whether alternative imputation strategy should be considered. I would suggest considering a tipping point analysis to check whether the results are consistent under different missing data assumptions (see for example https://pubmed.ncbi.nlm.nih.gov/12483769/).

5. For the TMLE approach, please clarify the model used to analyse the primary outcome. i.e. did you use a logistic regression (assuming a binomial distribution and logit link) or a linear regression (assuming a normal distribution and identity link), or something else? If using a logistic regression, please clarify how absolute differences were derived from the logistic model including the method to obtain confidence intervals.

6. Please report the covariates included in the final model(s).

7. The paragraph on HIV testing contains repetition with the reasons for declining HIV testing reported twice. Please correct.

8. In Table 1, please report denominators (the number of participants with data available). According to the text on Page 10, only 100 participants underwent HIV testing; however, based on the percentages reported in Table 1, the denominator includes all randomised participants. I believe it would be more accurate to report the proportion with HIV only in those who underwent testing unless we can be confident that those not tested were HIV negative. This may apply to other baseline variables with missing data (hence the need to report denominators and potentially adjust percentage calculations).

9. When describing baseline characteristics, please comment on the (im)balance in baseline characteristics in the manuscript, mentioning the important imbalance in baseline grade of hypertension.

10. Please report SBP and DBP continuously in Table 1 i.e. showing means, standard deviations and quartiles.

11. Please clarify how the risk differences and 95% confidence intervals for the primary and secondary outcome results (cf. bottom of page 10 / top of page 11 + Table 2) were obtained. It reads as if these are raw differences (e.g. 77% - 51% = 26%) which is concerning given the important baseline differences in blood pressure between the two arms. As mentioned above in my first major comment, I would strongly recommend that all analyses are adjusted for baseline blood pressure as a continuous variable. This could be done using logistic regression with or without additional baseline covariates.

12. Please consider reporting p-values for the primary and secondary outcome results in the text and abstract.

13. It would be helpful to see blood pressure reported as continuous outcome at baseline, week 24 and week 48. A figure (e.g. boxplot) would be helpful - this could potentially be added as a second panel to Figure 2 or in the supplement.

14. I would strongly recommend an analysis of covariance for week 24 SBP adjusted for baseline SBP. This could potentially take the form of a longitudinal model if including both week 24 and week 48 measurements. It is not clear whether the SBP results reported in Table 2 are adjusted for baseline SBP (please consider adding a footnote to Table 2).

15. Figure 3. For more clarity, please consider including headers for each subgroup e.g. country, age, gender, etc… and indent the labels (e.g. Kenya and Uganda) within each subgroup header.

16. Figure 3. As is recommended practice for subgroup analyses, please report interaction p-values (the p-value associated with the interaction term between the subgroup variable and the treatment group) instead of the p-value of treatment effect within the subgroup.

17. I note that the subgroup analysis is conducted on hypertension control at 48 weeks. Given that the primary outcome is measured at 24 weeks, I believe it would be more logical to perform the subgroup analysis on the week 24 outcome. A subgroup analysis at 48 weeks could potentially be included in the supplemental results; however, I would suggest removing it entirely.

18. Please make sure to use the same legend across all figures. Figures S1 and S2 seem to have the legend reversed (i.e. grey for control in S1 vs grey for intervention in S2).

19. Please mention imbalances in blood pressure status at baseline as a limitation.

-Laurent Billot

Reviewer #4: This is a small, pilot study of facilitated telehealth hypertension care in rural primary care clinics in Kenya and Uganda. The investigators showed that community health worker facilitated telehealth improved hypertension control. This is a well-written manuscript and I have minor comments as follows:

1) The time for enrollment was from May to Nov 2022, can the authors include any outcome data of patients that achieved blood pressure control early in the study as a secondary analysis moving forward to understand the direct impact?

2) Can you report mean time to blood pressure control and whether this differed by severity of hypertension at enrollment?

3) Can you also report number of medications needed to achieve blood pressure control by severity of hypertension at enrollment?

4) All the patients living with HIV were treated and virally suppressed. Did you notice any differences in ability to achieve hypertension control among persons living with HIV compared with persons without HIV such as longer time to achieve control or more medications needed to achieve control?

5) Related to the above, were there any differences among participants with multiple co-morbidities versus not in achieving hypertension control?

Reviewer #5: The authors have produced a clear and well written manuscript describing the results of an RCT comparing fully in person HTN care to a combination of telehealth facilitated by CHWs and less frequent clinic follow-up. The results are impressive for the pilot and can inform potential options for managing individuals most at risk for complications. There were however some areas which were confusing to this reviewer and which need clarification. In addition, while this is indeed a promising model, there were significant resources needed including engaging existing CHWs, telehealth through study clinicians as examples and addressing how this can be scaled will be important

Methods:

The intervention is described as leveraging HIV care, and integrated, but the study seems to be done at PHC centers? A description of where and how HIV care is delivered and where the HTN was done

It is not explicitly described the expectation for regular visits for the control arm (assume every month?)

Were medications provided free to both arms or are these available free of charge regardless

Were stockouts an issue for the clinic-based care (versus the packaged weekly medications for CHW delivery)

Was the clinical visit for retention in care including a CHW one or only at the clinic? If the latter, how was this adjusted based on the frequency of expected clinic visits? Measuring adherence to visits comparing a CHW-facilitated telemedicine home visit versus a clinic visit should be made more clear in the text. Was there any outreach for no-shows in the control arm?

Results

A consort diagram to help a reader understand differences in acceptability and accessibility -how did the 80% who came to clinic and those enrolled differed from those with HTN dxed.

The close to half with previous diagnosis is an important statistic. Were differences in effectiveness seen based on previous diagnosis (or exploratory for baseline medications?)

The section on reasons for not HIV testing, results are repeated

Medication results are ? missing (and would like to know quality of care-were both arms on recommended medications?

Discussion

The discussion is well laid out but again has some areas where clarification is needed. The paragraph about integrating HIV and HTN care requires more explanation-was this HTN integrated into HIV care? Did people without HIV got to an HIV clinic? Since the main outcome was status neutral, this terminology is confusing unless you have Hiv 9-) people coming to an HIV/HTN integrated clinic? See comments above

The limitations should include the potential for the study clinicians to influence the effectiveness (although in theory should be equal in intervention and control). See above other areas where clarity is made (supply chain support, ?medication costs), leveraging an exisiting CHW system.

Minor

Methods: the sentence "WithinTMLE, we used Adaptive Pre-specification to select the optimal adjustment approach via

sample-splitting; candidate adjustment variables were age, sex, baseline hypertension severity, country, and unadjusted." Seems incomplete?

---

* Please upload any figures associated with your paper as individual TIF or EPS files with 300dpi resolution at resubmission; please read our figure guidelines for more information on our requirements: http://journals.plos.org/plosmedicine/s/figures. While revising your submission, please upload your figure files to the PACE digital diagnostic tool, https://pacev2.apexcovantage.com/. PACE helps ensure that figures meet PLOS requirements. To use PACE, you must first register as a user. Then, login and navigate to the UPLOAD tab, where you will find detailed instructions on how to use the tool. If you encounter any issues or have any questions when using PACE, please email us at PLOSMedicine@plos.org.

* ETHICS STATEMENTS: Please provide details of the consent and whether it was written or oral. Please also provide the approval number.

* FINANCIAL DISCLOSURES: The funding statement should include: specific grant numbers, initials of authors who received each award, URLs to sponsors’ websites. Also, please state whether any sponsors or funders (other than the named authors) played any role in study design, data collection and analysis, the decision to publish, or preparation of the manuscript. If they had no role in the research, include this sentence: “The funders had no role in study design, data collection and analysis, decision to publish, or preparation of the manuscript.”

* COMPETING INTERESTS: All authors must declare their relevant competing interests per the PLOS policy, which can be seen here: https://journals.plos.org/plosmedicine/s/competing-interests

For authors with ties to industry, please indicate whether any of the interests has a financial stake in the results of the current study.

* DATA AVAILABILITY: PLOS Medicine requires that the de-identified data underlying the specific results in a published article be made available, without restrictions on access, in a public repository or as Supporting Information at the time of article publication, provided it is legal and ethical to do so. Please see the policy at http://journals.plos.org/plosmedicine/s/data-availability

and FAQs at http://journals.plos.org/plosmedicine/s/data-availability#loc-faqs-for-data-policy

The Data Availability Statement (DAS) requires revision. For each data source used in your study:

FIGURES AND TABLES

SUPPLEMENTARY MATERIAL

REFERENCES

* Where website addresses are cited, please include the complete URL and specify the date of access (e.g. [accessed: 12/06/2024]).

STUDY TYPE-SPECIFIC REQUESTS

* PLOS Medicine requires that all trials be prospectively registered in one of registries recognized by WHO. Please ensure that study registration details are included in the Methods section.

* Please structure the Methods section using the following sub-headings: Study design and participants, Randomization and masking, Procedures, Outcomes, Statistical analysis.

* The following outcomes measures time to attaining hypertension control (secondary outcome), hypertension control at 48 weeks, moderate-to-severe hypertension (average BP ≥160/100 mmHg) at 24 and 48 weeks, mean systolic blood pressure at 24 and 48 weeks, structured survey at 24 and 48 weeks appear to differ between the submitted manuscript and the protocol. Please clarify and explain all discrepancies between the paper and protocol. If the outcomes were not prespecified in the protocol, please define them in the Methods (Outcomes section) as post hoc and explain why they were added. Post-hoc comparisons should be presented as hypothesis generating rather than conclusive.

* Please ensure that all prespecified outcomes (primary, secondary, and exploratory) are listed in the Methods/Outcomes section and indicate whether there are outcomes that are not presented in the current report.

* Please specify the dates (Month Day, Year) during which study enrollment and follow up occurred.

* Please include absolute numbers wherever you report percentages; eg, n/N (%)

* Please present the safety data for the study including numbers of specific events and whether or not adverse events are thought to be related to treatment. AEs should be reported in the abstract, per CONSORT and CONSORT-Harms.

* Please complete the CONSORT checklist (https://www.equator-network.org/reporting-guidelines/consort/) and ensure that all components of CONSORT are present in the manuscript, including how randomization was performed, allocation concealment, blinding of intervention, definition of lost to follow-up, power statement. When completing the checklist, please use section and paragraph numbers, rather than page numbers.

* Please report your abstract according to CONSORT for abstracts, following the PLOS Medicine abstract structure (Background, Methods and Findings, Conclusions) https://www.equator-network.org/reporting-guidelines/consort-abstracts/

* If your trial had to undergo important modifications in response to extenuating circumstances, please complete the CONSERVE-CONSORT checklist and provide in your Supporting Information; (https://www.equator-network.org/reporting-guidelines/guidelines-for-reporting-trial-protocols-and-completed-trials-modified-due-to-the-covid-19-pandemic-and-other-extenuating-circumstances-the-conserve-2021-statement/). When completing the checklist, please use section and paragraph numbers, rather than page numbers.

* In keeping with our commitment to Open Science, please include the study protocol document and analysis plan (including any amendments) as Supporting Information to be published with the manuscript if accepted.

* Please note that PLOS Medicine requires prospective, public registration of a data sharing plan (as part of mandatory clinical trials registration) for all clinical trials that began enrollment on or after January 1, 2019, in accordance with ICMJE requirements.

---

## [Decision Letter · Decision Letter 2]

Dear Dr. Hickey,

Thank you very much for re-submitting your manuscript "Randomized trial of Community Health Worker Facilitated Telehealth for Moderate-Severe Hypertension Care in Kenya and Uganda" (PMEDICINE-D-24-03678R2) for review by PLOS Medicine. Thank you for your patience while awaiting this decision.

I would also like to thank you for your detailed response to the reviewers' and editors’ comments. I have discussed the paper with my colleagues and the academic editor, and it has also been seen again by two of the original reviewers. The changes made to the paper were satisfactory to the reviewers and the academic editor. As such, we intend to accept the paper for publication, pending your attention to the editors' comments below in a further revision. When submitting your revised paper, please once again include a detailed point-by-point response to the editorial comments.

[LINK]

In revising the manuscript for further consideration here, please ensure you address the specific points made by each reviewer and the editors. In your rebuttal letter you should indicate your response to the reviewers' and editors' comments and the changes you have made in the manuscript. Please submit a clean version of the paper as the main article file. A version with changes marked must also be uploaded as a marked up manuscript file. Please also check the guidelines for revised papers at http://journals.plos.org/plosmedicine/s/revising-your-manuscript for any that apply to your paper.

We ask that you submit your revision within 1 week (May 06 2025). However, if this deadline is not feasible, please contact me by email, and we can discuss a suitable alternative.

Please do not hesitate to contact me directly with any questions (atosun@plos.org). If you reply directly to this message, please be sure to 'Reply All' so your message comes directly to my inbox.

We look forward to receiving the revised manuscript.

Sincerely,

Alexandra Tosun, PhD

Associate Editor 

PLOS Medicine

plosmedicine.org

Comments from Reviewers:

Reviewer #3: Thank you for thoroughly addressing my previous comments. I feel re-assured that the baseline differences in hypertension grade were adequately accounted for and that the results are robust to adjustments. It is also reassuring to see that, when reported continuously, baseline differences in SBP and DBP are in fact minimal. I have no further comment.

-Laurent Billot

Reviewer #5: The authors have done a thorough and masterful response to the many reviewers and as a result, the paper is ready for acceptance.

[LINK]

Requests from Editors:

GENERAL

* Please confirm that your title complies with to PLOS Medicine's style. Your title must be nondeclarative and not a question. It should begin with main concept if possible. "Effect of" should be used only if causality can be inferred, i.e., for an RCT. Please place the study design ("A randomized controlled trial," "A retrospective study," "A modelling study," etc.) in the subtitle (ie, after a colon).

* Please ensure that all abbreviations are defined at first use throughout the text (including statistical abbreviations). Please also check figures and tables.

* Please review your text for claims of novelty or primacy (e.g. 'for the first time', ‘novel’) and remove this language.

* Please check that any use of statistical terms (such as trend or significant) are supported by the data, and if not please remove them.

* Please ensure that tables and figures, including those in supplementary files, are appropriately referenced in the main text.

* Statistical reporting: Please revise throughout the manuscript, including tables and figures.

a) Please report statistical information as follows to improve clarity for the reader "22% (95% CI [13%,28%]; p</=)".

b) Please separate upper and lower bounds with commas instead of hyphens as the latter can be confused with reporting of negative values.

c) Please define statistical definitions at first use and repeat the abbreviated definitions (HR, CI etc.) for each set of parentheses.

* Thank you for providing a protocol summary. Please note that we still require you to provide the full protocol as a Supporting Information file.

ABSTRACT

* Please confirm that your abstract complies with our requirements, including providing all the information relevant to this study type https://journals.plos.org/plosmedicine/s/submission-guidelines#loc-abstract

* Per CONSORT, please note that only the primary outcome of the trial should be reported in your Abstract. Secondary outcomes should only be included in the Abstract if all secondary outcomes are fully reported. For trials that have many secondary outcomes, the Abstract should be limited to reporting the primary outcome.

* Please ensure that all numbers presented in the abstract are present and identical to numbers presented in the main manuscript text.

* In the abstract, please include the important dependent variables that are adjusted for in the analyses.

* When reporting age, please ensure to include a unit such as years. Also, please repeat units, e.g. for BP (e.g. l.36, 37).

* Please specify who was blinded to the intervention and control.

* Please define the intervention and control states.

* Please state that analysis was intention to treat.

* Please provide the number of participants lost to follow up in each group.

AUTHOR SUMMARY

* In the final bullet point of 'What Do These Findings Mean?', please include the main limitations of the study in non-technical language.

METHODS AND RESULTS

* In the flow diagram, please indicate the number of individuals in each group analyzed in the ITT analysis.

* Causal language - In trials, there is usually a distinction in the language in terms of causal vs associational for primary and secondary trial outcomes. It would be beneficial to use associational language in the discussion and other sections for secondary outcomes.

* The terms gender and sex are not interchangeable (as discussed in https://www.who.int/health-topics/gender#tab=tab_1 ); please use the appropriate term.

* Figure 2: Please convert any stacked bar charts to another data representation for example a table, or other type of graph. Please ensure to provide the nominators and denominators for each group per time point.

* ll.426-429: Could you please clarify where, i.e. in a table or figure, these results can be found?

* Supplement Figures E-F: Does Clinic Factor equate to Quality of Care? If so, we think Quality of Care in Clinic would be a better description for the figures than Clinic Factor. We also suggest using the same description between the figures and the main text.

* l.453: Is there a specific reason why you chose to show the numbers from 180 days instead of, say, 200 days, which is on the x-axis?

* ll.458-464: Please ensure to provide a reference to the relevant table(s) and/or figure(s).

* ll.570-472: Could you please clarify where, i.e. in a table or figure, these results can be found?

DISCUSSION

* Please remove the 'conclusions' subheading.

General Editorial Requests

---

## [Editor Report · Decision Letter 3]

Dear Dr Hickey, 

On behalf of my colleagues and the Academic Editor, David Flood, I am pleased to inform you that we have agreed to publish your manuscript "Community Health Worker Facilitated Telehealth for Moderate-Severe Hypertension Care in Kenya and Uganda: A randomized controlled trial" (PMEDICINE-D-24-03678R3) in PLOS Medicine.

I appreciate your thorough responses to the reviewers' and editors' comments throughout the editorial process. We look forward to publishing your manuscript, and editorially there are only a few remaining points that should be addressed prior to publication. We will carefully check whether the changes have been made. If you have any questions or concerns regarding these final requests, please feel free to contact me at atosun@plos.org.

Please see below the minor points that we request you respond to:

* “If the Editor prefers that these be available in a figure, we can add an additional supplemental figure that includes the full forest plot of pre-specified sub-groups in which we analyzed our retention outcomes.” – We believe this would be a valuable addition to the manuscript and ask that you include the full forest plot as suggested. Please be sure to include a reference in the main text.

* ll.355-356 (marked up version): Please remove the funding statement from the main text. This information is only needed in the metadata of the online submission form.

* S1 Supplement, Figure D: Please note that the x-axis still shows days instead of weeks. Please revise.

Before your manuscript can be formally accepted you will need to complete some formatting changes, which you will receive in a follow up email (including the editorial points above). Please be aware that it may take several days for you to receive this email; during this time no action is required by you. Once you have received these formatting requests, please note that your manuscript will not be scheduled for publication until you have made the required changes.

PRESS

Sincerely, 

Alexandra Tosun, PhD 

Associate Editor 

PLOS Medicine